# Plant Metabolites as Potential Agents That Potentiate or Block Resistance Mechanisms Involving β-Lactamases and Efflux Pumps

**DOI:** 10.3390/ijms26125550

**Published:** 2025-06-10

**Authors:** Muhammad Jawad Zai, Ian Edwin Cock, Matthew James Cheesman

**Affiliations:** 1Centre for Planetary Health and Food Security, Griffith University, Brisbane, QLD 4111, Australia; muhammadjawad.zai@griffithuni.edu.au; 2School of Environment and Science, Griffith University, Brisbane, QLD 4111, Australia; 3School of Pharmacy and Medical Sciences, Griffith University, Southport, QLD 4222, Australia

**Keywords:** phytochemicals, efflux pump inhibitor, β-lactamase inhibitor, antibiotic resistance, combinational therapies, antibiotic potentiation

## Abstract

The dramatic increase in antimicrobial resistance (AMR) in recent decades has created an urgent need to develop new antimicrobial agents and compounds that can modify and/or block bacterial resistance mechanisms. An understanding of these resistance mechanisms and how to overcome them would substantially assist in the development of new antibiotic chemotherapies. Bacteria may develop AMR through multiple differing mechanisms, including modification of the antibiotic target site, limitation of antibiotic uptake, active efflux of the antibiotic, and via direct modification and inactivation of the antibiotic. Of these, efflux pumps and the production of β-lactamases are the most common resistance mechanisms that render antibiotics inactive. The development of resistance-modifying agents (particularly those targeting efflux pumps and β-lactamase enzymes) is an important consideration to counteract the spread of AMR. This strategy may repurpose existing antibiotics by blocking bacterial resistance mechanisms, thereby increasing the efficacy of the antibiotic compounds. This review focuses on known phytochemicals that possess efflux pump inhibitory and/or β-lactamase inhibitory activities. The interaction of phytochemicals possessing efflux pumps and/or β-lactamase inhibitory activities in combination with clinical antibiotics is also discussed. Additionally, the challenges associated with further development of these phytochemicals as potentiating agents is discussed to highlight their therapeutic potential, and to guide future research.

## 1. Introduction

Bacterial resistance to clinical antibiotics (AMR) poses a global challenge, contributing significantly to high mortality and morbidity [1]. The increased incidence of multidrug resistance (MDR) in both Gram-negative and Gram-positive bacteria has led to infections that are challenging (or impossible) to treat with current antibacterial therapies [2]. The rise in AMR has been driven by multiple factors, including the increased use of antimicrobials in both humans and animals, and inappropriate prescribing of these therapies (e.g., to treat viral infections). Excessive and improper antibiotic use in humans promotes the emergence of resistant microorganisms [3]. The Centres for Disease Control and Prevention (CDC) reported that over 2 million people in the United States (U.S.) fall ill each year due to antimicrobial-resistant infections, leading to more than 23,000 deaths annually [4]. The cost associated with these resistant infections is estimated to be from USD7000 to over USD29,000 per patient [5]. Indeed, methicillin-resistant *Staphylococcus aureus* (MRSA) infections cost over USD18,000 each to treat effectively in the U.S., nearly EUR9000 in Germany, and more than 100,000 Swiss francs per case in Switzerland [6,7,8].

Bacterial antibiotic resistance has arisen from multiple origins, with various adaptive mechanisms identified that contribute to resistant behaviour in bacterial populations [9]. Efflux pumps are perhaps the quickest and most effective resistance mechanism bacteria can develop to neutralise antibiotics. Indeed, MDR efflux pumps are a major factor in the reduced susceptibility to antimicrobials seen in multiple clinical pathogenic bacteria isolates [10]. To date, six families of bacterial drug efflux pumps have been identified that contribute to efflux pathways [11]. The ATP binding cassette (ABC) family of efflux pumps directly uses ATP as an energy source to power antimicrobial movement out of the cell [12]. The other five groups, which include the multidrug and toxin extrusion family, major facilitator family, small multidrug resistance family, proteobacterial antimicrobial compound efflux family, and the resistance nodulation cell division superfamily function as secondary active transporters, with efflux driven by electrochemical energy from transmembrane ion gradients [13,14].

Antibiotics of the β-lactam class are among the most prescribed antibiotics for treating microbial infections. They are valued for their safety, reliable (and often broad-spectrum) bactericidal properties, and their clinical effectiveness. However, β-lactam antibiotic effectiveness is increasingly compromised by the global spread of genes encoding β-lactamase enzymes with extensive hydrolytic abilities, particularly in MDR Gram-negative pathogens. β-Lactamase enzymes are categorised into four classes based on their sequence identities [15]. Whilst only a few β-lactamase enzymes had been documented by the early 1970s, their numbers have since grown rapidly, with new enzymes capable of hydrolysing carbapenems now increasingly identified in clinical isolates [16]. A notable example is the class A β-lactamase KPC-2 which, within a few years, has become one of the most concerning β-lactamases spreading globally [17]. Class A β-lactamases may be plasmid-encoded (GES, KPC, FRI-1), chromosomally encoded (NmcA, SME, BIC-1, SFC-1, FPH-1, PenA, SHV-38), or both (IMI). Plasmid encoded enzymes are frequently linked to mobile genetic elements that facilitate their transfer. These enzymes exhibit shared structural characteristics and operate through a common mechanism of action. They variably hydrolyse cephalosporins, penicillins, carbapenems, monobactams, and are inhibited by tazobactam and clavulanate [17].

In recent years, there has been growing interest in natural products for the development of new antibiotic chemotherapies, as numerous studies have shown that natural product compounds exhibit significant antimicrobial effects, which are often distinct from those of existing clinical antibiotics [18,19]. Notably, phytochemicals that exert antimicrobial activity via mechanism(s) that differ from those of presently available antibiotics may be of substantial clinical value for treating antibiotic-resistant infections [20,21]. Many phytochemicals have demonstrated effective antibiotic resistance reversal activity, primarily by inhibiting drug efflux pumps or modifying enzymes [22].

## 2. Medicinal Plant Secondary Metabolites as Antimicrobial Agents

Plants derive their antimicrobial properties from their secondary metabolites. Plant secondary metabolites constitute over 8000 known phenolic compounds, of which there are more than 12,000 known alkaloids and 25,000 terpenoids, many of which possess strong antibacterial activity [23]. Of these, phenolic compounds are the most frequently reported antibacterial agents [24]. Phenolic compounds can contain either multiple phenol units, in which case they are classified as polyphenolic, or they may contain a single substituted phenolic ring. Phenols can be further classified into phenolic acids, coumarins, stilbenes, flavonoids, lignans, and tannins (Figure 1). In particular, terpenes and terpenoids are widely recognised for their antimicrobial activities [25]. Another noteworthy class of plant-derived antimicrobials are alkaloids. These compounds form a diverse and structurally varied group, serving as foundational scaffolds for critical antibacterial drugs including quinolones and metronidazole [26]. The antibacterial activity of selected plant secondary metabolites is summarised in Table 1.

## 3. Bacterial Efflux Pump Mechanisms

Antibiotic efflux is one of the most prevalent resistance mechanisms among various pathogenic bacteria [46]. Efflux pumps play a role in maintaining the internal environment of bacteria by expelling quorum-sensing molecules, toxic substances, bacterial virulence factors, and biofilm-forming molecules [47]. Efflux pumps can be selective for a single substrate, or capable of transporting a variety of structurally diverse compounds, including multiple classes of antibiotics [48]. Efflux pumps are classified as primary or secondary pumps, based on the energy source they use to expel substrates. Primary efflux pumps use the energy derived from ATP hydrolysis to transport substrates across the membrane. In contrast, secondary efflux pumps rely on energy from electrochemical gradients, such as the proton motive force, created by protons or ions. To date, six main families of bacterial efflux pumps have been identified, and include the major facilitator superfamily, the ATP binding cassette superfamily, the resistance nodulation cell division family, the multidrug and toxic compound extrusion family, the proteobacterial antimicrobial compound efflux and the small multidrug resistance family (Figure 2) [12]. Efflux pumps are crucial for bacterial survival in various stress environments, making them a promising target for developing new inhibitors to restore the effectiveness of existing antibiotics.

## 4. Mechanism of β-Lactamases

The extensive use of β-lactam antibiotics has driven the emergence and spread of resistance. Enzyme-mediated β-lactam resistance is due to the action of the β-lactamase enzymes, which are produced by both Gram-negative and Gram-positive bacteria. The enzyme hydrolyses the β-lactam amide bond (Figure 3) [49]. β-Lactamases are categorised into the four classes A, B, C, and D and this is based on primary sequence homology and variations in their hydrolytic mechanisms [50]. Class A, C, and D β-lactamases are serine-based enzymes that hydrolyse the β-lactam ring through a serine-bound acyl intermediate in the active site. In contrast, class B β-lactamases, known as metallo-β-lactamases, contain one or two zinc ions in their active site, which are essential for their enzymatic function [51]. Over 400 distinct types of β-lactamases have been identified. Among the Gram-negative bacteria, including *Enterobacteriaceae* spp., *Haemophilus influenza*, *P. aeruginosa*, and *Neisseria gonorrhoeae*, the most common plasmid-mediated class A β-lactamases are TEM and a closely related enzyme, TEM-2. These enzymes are capable of hydrolysing multiple penicillins, and some cephalosporins [52]. A related, but less prevalent class A β-lactamases called SHV have been identified primarily in *Enterobacteriaceae* spp. and in *P. aeruginosa*. These enzymes can hydrolyse aztreonam and third generation cephalosporins, including ceftazidime and cefotaxime [52].

## 5. Phytochemicals as Efflux Pump and β-Lactamase Inhibitors

Herbal remedies and phytochemical-based therapies have gained substantial recent attention as targets for the discovery of efflux pumps and β-lactamase inhibitors due to their structural diversity and multiple modes of action. Berberine (Figure 4a) was first discovered by Herberger and Buchner in 1830 and is the most extensively researched protoberberine alkaloid found in nature, with a long history of use [53]. The primary plants containing berberine include *Berberis vulgaris* L., and *Scutellaria baicalensis* Georgi., which have been traditionally used as folk medicines in India, China, Iran, and various other countries [54]. Berberine’s pharmacological activity has been associated with nearly all types of bodily disorders, including blood disease and cancer [55,56], cardiovascular diseases [57], the central nervous system [58] and immune diseases [59]. Research on medicinal plant extracts has revealed several phytochemicals that exert antimicrobial activity by inhibiting efflux pumps and β-lactamases in Gram-positive and Gram-negative bacteria (Table 2). By blocking these resistance mechanisms, such phytochemicals enable antibiotics to reach sufficient concentrations within bacterial cells to achieve bactericidal effects.

## 6. Plant Metabolite Structure Activity Relationship

Plant metabolites are known for their wide range of biological activities, including antimicrobial effects and their extensive structural diversity makes them promising candidates for drug discovery [90]. The most common classification of phytochemicals is based on their chemical structures, including groups such as alkaloids, limonoids, phenolics, saponins, secoiridoids, terpenoids and others. Alkaloids are typically defined as nitrogen-containing basic metabolites with heterocyclic structures, although this definition does not always clearly distinguish them from other nitrogenous compounds. They are classified in various ways, often based on their biogenic origins or the characteristics of their carbon skeleton. Among phytochemicals, alkaloids exhibit particularly high structural diversity. They are generally categorised according to their core carbon skeletons [91]. Limonoids are distinctive secondary metabolites characterised by a tetranortriterpenoid backbone that includes a furan ring. They are commonly isolated from *Citrus* and *Meliaceae* plants and have been shown to exhibit antibacterial activities [92]. Several simple phenols and phenolic acids also exhibit antibacterial activities against a broad range of pathogens. Structure-activity relationship studies suggest that phenols with varying alkyl chain lengths and hydroxyl groups may play a crucial role in determining antimicrobial activity [93]. Studies have reported that the antimicrobial activity of phenolic acid derivatives increases with the length of the alkyl chain [94]. The antimicrobial toxicity of phenolic compounds is also influenced by the presence of hydroxyl groups and the oxidation state of the phenol moiety. An increase in the length of hydrophobic alkyl chains can disrupt the fluidity of microbial cell membranes. Phenolic acids may integrate into the membrane structure, with their polar hydroxyl groups forming hydrogen bonds in the aqueous phase, while their nonpolar alkyl chains align with the lipid bilayer through dispersion forces [93]. Therefore, when the hydrophilic interactions outweigh the hydrophobic ones, the antimicrobial activity tends to diminish. Additionally, the number and position of substituents on the benzene ring, along with the length of the saturated side chain, influence the bactericidal effects of phenolic acids. However, these effects vary between Gram-positive and Gram-negative bacteria [95].

## 7. Synergistic Interactions Between Efflux Pump and β-Lactamase Inhibitor Phytochemicals and Conventional Antibiotics

Synergistic interactions refer to a phenomenon where the combined therapeutic potential of two or more compounds is greater than the sum of their individual effects. Combinations of phytochemicals possessing efflux pump and/or β-lactamase inhibitor activity with conventional antibiotics (or antibiotic natural compounds) can reduce the minimum inhibitory concentration (MIC) of the antibiotic components and thus sensitise MDR bacteria to these antibiotics [96]. In synergistic interactions, one molecule can enhance or support the primary action of another compound, resulting in improved therapeutic efficacy. Such synergistic interaction occurs when antibiotics are paired with compounds capable of partially or entirely inhibiting bacterial resistance mechanisms. These combinations enhance the effectiveness of antibiotics by reducing bacterial defences, allowing for a more robust therapeutic response. For example, combining clavulanic acid with penicillin (or other β-lactam antibiotics) effectively blocks penicillinase resistance [97]. Table 3 summarises known combinations of phytochemicals and antibiotics that minimise antimicrobial resistance by targeting efflux pumps and β-lactamase enzymes.

## 8. Plant Metabolites Efflux Pump Inhibitor Mechanisms

Efflux pump inhibitors aim to restore bacterial sensitivity to antibiotics by preventing the bacteria from expelling the drug through their intramembrane efflux pump systems [119]. Efflux pump inhibitors work through various methods, including blocking of ATP synthesis to inhibit antibiotic efflux, repressing the gene responsible for encoding efflux pumps and dissipation of the proton gradient essential for the pump activity. Other mechanisms include disrupting the assembly of efflux systems and obstructing the membrane proteins involved in efflux (Figure 5).

Different studies have examined baicalin for its ability to inhibit the MsrA efflux pump in ARSS (azithromycin-resistant *Staphylococcus saprophyticus*), a strain that expresses the MsrA efflux gene and is resistant to azithromycin [120]. MsrA is an ATP-dependent efflux pump, suggesting that baicalin’s inhibitory effects on the ARSS efflux pump are more closely associated with the ATP-dependent processes in bacteria [121]. Baicalin may disrupt ATP production, potentially inhibiting the function of the MsrA efflux pump and enhancing its efflux pump inhibitory activity. Studies have also shown capsaicin (Figure 4f) is an effective efflux pump inhibitor activity against *S. aureus* strains that express NorA efflux pump [70]. Capsaicin is a known P-glycoprotein (P-gp) inhibitor, an active drug transporter belonging to the ATP-binding cassette transporter family. P-gp is involved in the efflux of chemotherapeutic agents from the cells and is highly expressed in the apical membrane of various pharmacologically significant epithelial barriers, such as the hepatocytes, intestinal epithelium and renal tubular cells indicating its critical role in drug absorption, distribution and elimination [122,123].

Cumin, and its key active component cuminaldehyde, inhibit the activity of the *S. aureus* LmrS multidrug efflux pump [76]. Host cells lacking the LmrS efflux pump exhibited a reduced response to treatment with cumin extract. Studies have also noted significantly higher ethidium bromide (a common substrate for efflux pump assays) accumulation in *E. coli* KAM32 host cells expressing the LmrS multidrug efflux pump when treated with cumin extract compared to KAM32 host cells lacking the LmrS efflux pump [76]. These efflux and accumulation studies suggest that cumin extract specifically targets the LmrS multidrug efflux pump. Cumin and cuminaldehyde also suppress the growth of host cells lacking LmrS, albeit at relatively higher concentrations [76]. This observation suggests that these compounds may disrupt the proton motive force generated during respiration and metabolism. Cumin appears to exert a multifaceted effect, inhibiting the LmrS efflux pump at low concentrations and compromising the integrity of the cell membrane at higher concentrations.

The plant alkaloid reserpine directly binds specific RND and MFS efflux pumps in Gram-negative bacteria through competitive inhibition [119]. In Gram-positive bacteria, reserpine interacts directly with specific amino acid residues within the efflux pump transporter proteins, such as the Bmr efflux pump in *B. subtilis* [119]. This interaction affects the pump ability to expel antimicrobial agents, thereby reducing resistance and potentially making the bacteria more susceptible to treatment with the antibiotic. Eugenol, an active component of clove oil and *trans*-cinnamaldehyde extracted from cinnamon bark inhibit efflux pumps in *A. baumannii* at a genetic level by downregulating gene *ade*A and *ade*B in the RND efflux family [124].

Catechin gallates, including epicatechin gallate and epigallocatechin gallate (Figure 4i), are plant-derived flavonoid-tannin hybrids that act as mild inhibitors of the NorA efflux pump, with epicatechin gallate showing slightly greater potency [119]. It has been suggested that these molecules interact with two distinct binding sites on the NorA efflux transporter, each with varying affinities. At low concentrations, catechins bind to high-affinity sites, increasing efflux of the NorA substrate [119]. Epigallocatechin gallate also acts on TetK efflux pumps in *Staphylococci* spp. and *Campylobacter* spp. to increase their susceptibility to tetracycline, erythromycin and ciprofloxacin, although the exact mechanism was not determined [119]. In *K. pneumoniae*, epigallocatechin has been shown to restore susceptibility to tetracycline due to a change in the capsule structure of *K. pneumoniae* and the inhibition of efflux pumps from the RND family (AcrAB) [125]. Epigallocatechin gallate also inhibits the growth of *B. subtilis* by targeting cell surface proteins including glucose phosphotransferase system transporter protein, oligopeptide ABC transporter binding lipoprotein, penicillin-binding protein 5 and phosphate ABC transporter substrate-binding protein [126].

Resveratrol, a phytochemical sourced from the seeds and skins of red grape, is another efflux pump inhibitor [127]. Resveratrol has been shown to enhance the susceptibility of clinical *A. baumannii* isolates resistant to chlorhexidine [127]. Whilst resveratrol itself does not inhibit the growth of *A. baumannii*, it targets the expression of AdeB protein, a component of AdeABC efflux pump system. The expression of AdeB is significantly reduced when resveratrol is combined with chlorhexidine, thereby substantially increasing antibiotic sensitivity [127].

## 9. Plant Metabolites β-Lactamase Inhibitor Mechanism

β-Lactam antibiotics are among the most used antimicrobial agents in hospital environments due to their broad spectrum of activity, high efficacy and safety [128]. The four primary classes of β-lactam antibiotics include cephalosporins, penicillins, monobactams and carbapenems, all of which feature a four-membered cyclic amide (azetidinone), commonly known as β-lactam rings. All β-lactam antibiotics share a common mode of bacterial killing, via the inhibition of transpeptidase enzymes that are essential for the final step in bacterial cell wall synthesis. Alexander Fleming observed that the β-lactam antibiotic penicillin failed to inhibit the growth of certain bacteria within the colityphoid group [129]. Further studies revealed an enzyme in *E. coli* (initially termed as penicillinase) inactivated penicillin by breaking its β-lactam ring, producing penicilloic acid, thereby rendering it ineffective [130]. One strategy to counteract bacterial resistance to β-lactam antibiotics involves the targeting and inhibition of the β-lactamase enzyme (Figure 6). Clavulanic acid, a naturally occurring compound derived from *Streptomyces clavuligerus*, has been identified as a powerful inhibitor of staphylococcal and TEM-1 penicillinase and is perhaps the most widely known example of a β-lactamase inhibitor [131]. Mechanistically, clavulanic acid is a high-efficacy irreversible inactivator of TEM-1, creating an irreversibly inactivated enzyme complex [132].

The phytochemical 2-methoxy chrysophanol (Figure 4c) derived from *Clutia myricoides* Jaub. & Spach possesses an anti-ESBL activity against *K. pneumoniae*. To investigate the mechanism of action of 2-methoxy chrysophanol, molecular docking was conducted with CTX-M-27 ESBL [61]. 2-Methoxy chrysophanol formed two hydrogen bonds with the side chains of Ser70 and Thr235 through its two phenolic hydroxyl groups. Although it did not form a hydrogen bond with the catalytic residue Ser237, its carbonyl group compensated by forming a hydrogen bond with the nitrogen atom of the backbone amide of this residue. Notably, 2-methoxy chrysophanol also established two critical hydrogen bonds via its methoxy group with the side chains of Asn132 and Asn104. These interactions are particularly significant as they contribute to the compound’s activity against ESBL enzymes [61].

Studies have reported the β-lactamase inhibitory effects of 1,4-naphthalenedione (Figure 4b), a compound isolated from the plant *Holoptelea integrifolia* (Roxb.) Planch [60]. Molecular docking studies showed that 1,4-naphthalenedione binds to the active site of β-lactamase and one of its carbonyl oxygens hydrogen bonds with residues Lys 73 and Ser 70. The ligand is surrounded by several hydrophobic residues, suggesting that van der Waals interactions may serve as the primary stabilising factor of the complex [60].

Epigallocatechin, a phytochemical present in green tea, also is an effective inhibitor of β-lactamase enzymes. It binds directly to peptidoglycan in the bacterial cell membrane and inhibits penicillinase activity in a dose-dependent manner, achieving 50% inhibition at a concentration of 10 µg/mL [133]. The synergistic interaction between quercetin and amoxicillin has also been shown to be very effective in inhibiting peptidoglycan synthesis in the bacterial cell membrane, enhancing cell membrane permeability, suppressing β-lactamase activity, reducing the presence of fatty acids within bacterial cells and increasing protein amide I (alpha-helix and beta-sheet) and amide II levels [134]. These findings suggest that quercetin, both individually and in combination with amoxicillin, may alter the fatty acid composition of various membrane amphiphiles, resulting in cytoplasmic membrane damage and increased membrane permeability. Additionally, this combination may interact with β-lactamase, transpeptidase and other proteins, potentially forming complexes that lead to the stacking and accumulation of proteins within the bacterial cells.

## 10. Challenges and Future Perspectives

Some phytochemicals offer natural antibacterial properties and are generally safer than synthetic alternatives. Additionally, other compounds have antibiotic-potentiating activity by blocking bacterial resistance pathways. Herbal material containing bioactive phytochemicals is often readily available and locally accessible, easy to administer, cost-effective and provides significant therapeutic benefits, making treatments more affordable and practical for many communities [135]. The current proportion of antibacterial agents derived from medicinal plants does not represent their full potential for future antimicrobial therapies [136]. Isolating individual phytochemicals with targeted antibacterial activity can be a lengthy process and may require the use of large quantities of plant material and extraction solvents that can be toxic or environmentally damaging. Synergistic interactions within complex mixtures of compounds pose specific challenges, as current technology is not yet fully equipped to analyse multiple compounds working together across various biological targets [137]. The concept of synergistic effects is not standardised due to differences between the studies, and there is no uniform use of the term across studies. The interactions between phytochemicals can be favourable, such as synergy, or harmful, antagonistic. Synergistic interactions among medicinal plant extracts, as well as between their compounds and antibiotics, should be further investigated to uncover the underlying mechanisms driving their antimicrobial activity. This research could help identify multiple pathways that can be targeted for enhanced therapeutic outcomes. Phytochemicals are a largely untapped reservoir of bioactive compounds, with only a limited number of them having been explored for the discovery of new compounds that can modify AMR and serve as an effective therapeutic tool. Whilst all of the combinational therapies summarised herein highlight the potential to overcome bacterial resistance mechanisms, it is noteworthy that the bacteria may develop resistance against the potentiating compound, and further work is required to address this.

Future in vitro studies should assess the clinical relevance of these compounds and validate their in vivo efficacy through established correlations with in vitro efficacy results. Further studies should identify efflux pump and β-lactamase inhibitors by performing further studies including combinational, metabolomics and in silico methods. The structures of these compounds should also be modified to optimise their pharmacodynamics and pharmacokinetics, alongside conducting structure-activity analysis to enhance their safety and efficacy. Studies should also focus on investigating synergistic interactions within phytochemicals and antibiotics, to elucidate mechanisms underlying their antibacterial activity. This could reveal multiple molecular pathways to target, broadening the therapeutic options available to counter AMR. The interactions between phytochemicals and antibiotics can be either beneficial such as synergistic, or unsafe, as in antagonistic. Thus, additional studies are essential, particularly in vivo and toxicity studies, for these products to be validated and recognised as antimicrobial agents

## 11. Conclusions

The increasing incidence of antibiotic-resistant infections poses a challenge to effective bacterial infection treatment and control. Developing new and effective treatments against antibiotic-resistant pathogens has become a global priority, as these alternatives are essential to address the growing challenge of AMR. Whilst there is a relatively good amount of data reporting synthetic inhibitors, plant-based compounds are relatively ignored. Phytochemicals have the potential as a promising alternative to conventional antibiotics for combating infections caused by antibiotic-resistant pathogens. There are multiple other mechanisms that render the antibiotics ineffective. In this review, we focus on only two resistance mechanisms: efflux pumps and β-lactamases. These are arguably the most important because β-lactams are widely used, and efflux pumps are a common resistant mechanism. Some phytochemicals can inhibit key antibiotic resistance mechanisms, including efflux pumps and β-lactamase activity, which are necessary for pathogen survival and resistance. Although there are numerous examples of phytochemicals effectively combating antibiotic-resistant infections, the success in translating these findings into commercial applications remains limited and requires substantial improvement. There is a critical need to accelerate research, clinical trials and regulatory approval processes to facilitate the application of phytochemicals in combating MDR infections.

## Figures and Tables

**Figure 1 ijms-26-05550-f001:**
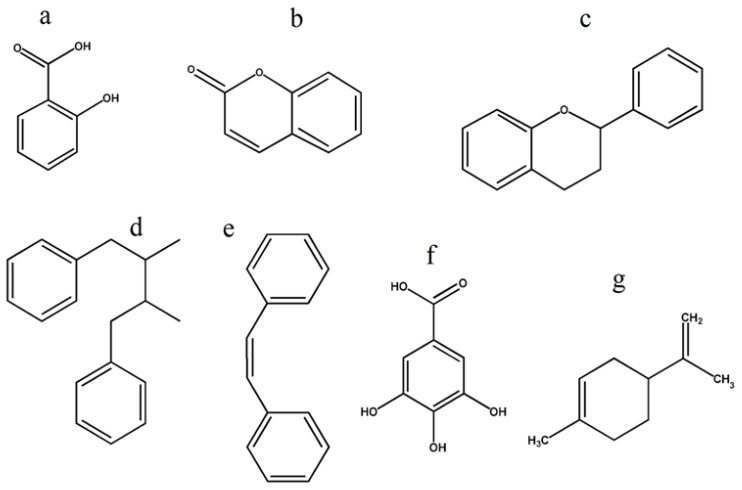
Base structures of (**a**) phenolic acid, (**b**) coumarins, (**c**) flavonoids, (**d**) lignans, (**e**) stilbenes, (**f**) tannins and (**g**) terpenoids.

**Figure 2 ijms-26-05550-f002:**
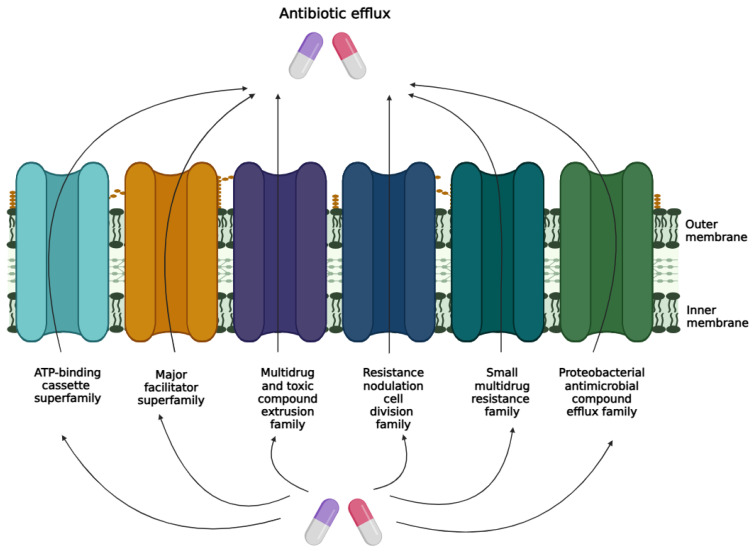
Efflux pump families.

**Figure 3 ijms-26-05550-f003:**
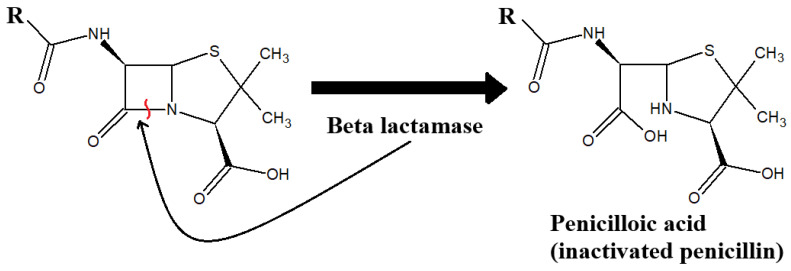
Inactivation of penicillin by beta lactamase enzymes. The red line indicates the peptide bond that is hydrolysed by the enzyme.

**Figure 4 ijms-26-05550-f004:**
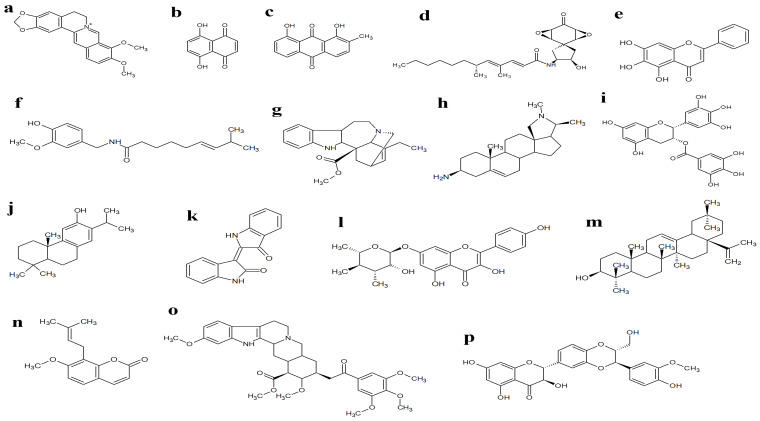
(**a**) Berberine, (**b**) 1, 4-naphthalenedione, (**c**) 2-methoxy chrysophanol (**d**) aranorosin, (**e**) baicalein, (**f**) capsaicin, (**g**) catharanthine, (**h**) conessine, (**i**) epigallocatechin gallate, (**j**) ferruginol, (**k**) indirubin, (**l**) kaempferol rhamnoside, (**m**) oleanolic acid, (**n**) osthol, (**o**) reserpine and (**p**) silybin.

**Figure 5 ijms-26-05550-f005:**
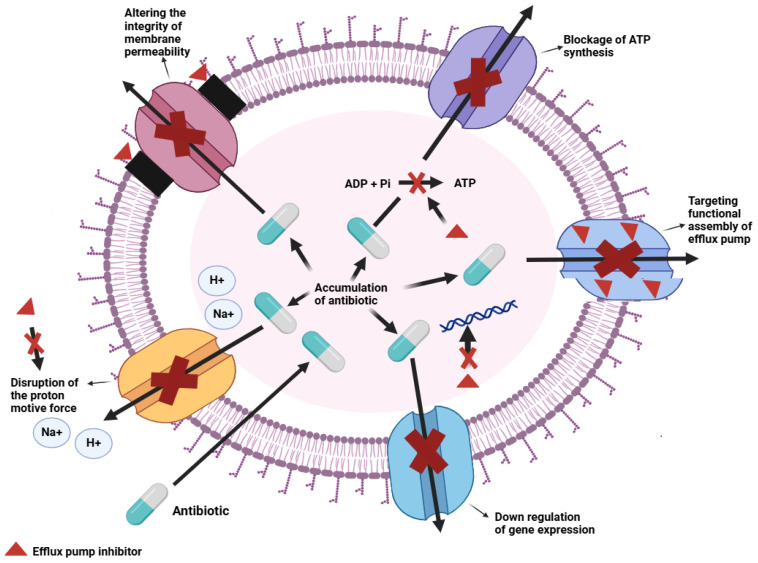
Efflux pump inhibitory mechanisms (indicated by the red crosses), including proton motive force disruption, blockage of ATP synthesis, altering the integrity of the membrane permeability, down-regulating gene expression and targeting the functional assembly of the efflux pump. The black rectangles indicate altered membrane permeability.

**Figure 6 ijms-26-05550-f006:**
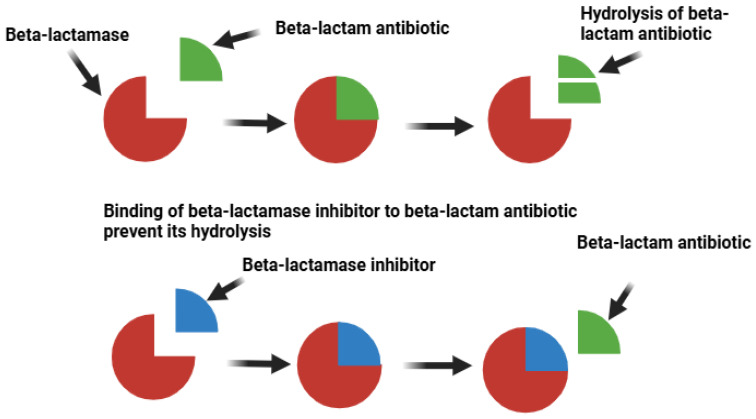
β-lactamase inhibitor mechanism of action.

**Table 1 ijms-26-05550-t001:** Antimicrobial activity of plant secondary metabolites.

Plant Name	Secondary Metabolites	Antimicrobial Activity Against	References
*Arbutus unedo* L.	Anthocyanins, flavonoids, Quinines, tannins	*S. aureus*, *E. coli*, *P. aeruginosa*	[27]
*Artabotrys crassifolius* Hook.f. & Thomson	Liridine	*B. cereus*	[28]
*Bauhinia purpurea* Linn.	Alkaloids, fatty acids, phytol esters Steroids, triterpenoids,	*A. niger*, *B. subtilis*, *Claviceps purpurea*, *E. coli*, *Klebsiella*, *P. aeruginosa*, *S. aureus*, *S. typhimurium*	[29]
*Berberis vulgaris* L., *Berberis petiolaris* Kunth., *Berberis aristate* DC., *Berberis integerrima* Bonge.	Berberine	*Brucella abortus*	[30]
*Cananga odorata* (Lam.) Hook. f. & Thomson.	Sampangine	*M. intracellulare*	[31]
*Cannabis sativa* L.	Cannabinoids	MRSA, *E. faecalis*, *S. pneumoniae*	[32]
*Cassia auriculatalinn* Linn.	Alkaloids, flavonoids, glycosides, phenols terpenoids, proteins, saponins, tannins,	*B. subtilis*, *C. albicans*, *Aspergillus niger*, *E. coli*, *S. aureus*	[33]
*Cnidoscolus acontifolius* (Mill) I.M.Johnst, *Newbouldia laevis* (P.Beauv.) Seem.ex Bureau, *Adansonia digitata* L., *Alchornea laxiflorab* (Benth.) Pax & K. Hoffm.,	Alkaloids, anthraquinones, reducing sugars flavonoids, phenols, resins, Saponins, steroids tannins, terpenoids, carbohydrates, cardioactive glycosides	*E. coli*, *S. aureus*, *B. subtilis*, *P. aeruginosa*	[34]
*Gloriosa superba* Linn.	Terpenoids, tannin, steroids, glycosides, alkaloids	*B. cereus*, *B. subtilis*, *E. coli*, *K. pneumonia*, *P. aeruginosa*, *P. vulgaris*, *S. aureus*, *S. faecalis*, *S. typhimurium*, *S. cremoris*	[35]
*Glycyrrhiza inflata* Batalin.	Licochalcone A	*S. aureus*	[36]
*Mammea americana* L.	Mammea B/BA	MRSA, *S. aureus*	[37]
*Momordica charantia* L.	Alkaloids, flavonoids, glycosides, Saponins, steroids, tannins	*B. subtilis*, *E. coli*, *P. aeruginosa*, *S. aureus*	[38]
*Ocimum sanctum* L.	Alkaloids, Steroidal compounds, tannins	*E. coli*, *S. aureus*, *P. mirabilis*	[39]
*Phyllanthus emblica* L.	Alkaloids, fats, glyceroids, carbohydrates, phenolics, lignin, oil, tannins, flavonoids, saponins, terpenoids	*E. coli*, *S. aureus*, *P. aeruginosa*, *B. subtilis*	[40]
*Psidium guajava* L.	Alkaloid, polyphenols, saponins, tannins, terpenoids,	*S. aureus*, *E. coli*, *P. aeruginosa*	[41]
*Punica granatum* L.	Punicalagin	*S. aureus*, *P. aeruginosa*, *Enterococcus mundtii*	[42]
*Rhodomyrtus tomentosa* (Ait.) Hassk.	Rhodomyrtosone B	Vancomycin-resistant *E. faecium*, MRSA	[43]
*Sanguinaria canadensis* L.	Sanguinarine	*B. subtilis*, *S. epidermidis*, *S. aureus*, MRSA	[44]
*Schinus lentiscifolius* Marchand.	Moronic acid	*S. pyogenes*, *S. aureus*, *B. subtilis*	[45]

**Table 2 ijms-26-05550-t002:** Phytochemicals possessing efflux pump and β-lactamase inhibition activity.

Phytochemicals	Plant Name	Efflux Pump Inhibitory Activity Against	β-Lactamase Inhibitory Activity Against	References
1,4-Naphthalenedione (Figure 4b)	*Holoptelea integrifolia* (Roxb.) Planch.	Not reported	*S. aureus*	[60]
2-Methoxy chrysophanol (Figure 4c)	*Clutia myricoides* Jaub. & Spach	Not reported	*K. pneumoniae*	[61]
5-O-Methylglovanon	*Glycosmis* plants	Not reported	*S. epidermidis* (MIC = 25–50 µg/mL) *and* ampicillin-resistant *S. aureus* (MIC = 12.50–50 µg/mL)	[62]
Abietane diterpenes	*Rosmarinus officinalis* L.	*S. aureus* (MIC = 16–64 µg/mL)	Not reported	[63]
Alkaloid compounds	*Cienfuegosia digitata* Cav.	Not reported	Ampicillin and methicillin-resistant *S. aureus* (ZOI = 10–16 mm)	[64]
Aranorosin (Figure 4d)	*Gymnascella aurantiaca*. Peck	Not reported	MRSA (ZOI = 10 mm)	[65]
Baicalein (Figure 4e)	*Scutellaria baicalensis Georgi*	MRSA (MIC = 64–256 µg/mL)	*S. aureus* (MIC = 128 µg/mL)	[66,67]
Berberine (Figure 4a)	*Berberis* spp.	*P. aeruginosa* (MIC = 125–250 µg/mL)	Not reported	[68]
Caffeoylquinic acids	*Artemisia absinthium* L.	*E. faecalis*, *S. aureus* (MIC = 32–>256 µg/mL)	Not reported	[69]
Capsaicin (Figure 4f)	*Capsicum* spp.	*S. aureus* (MIC ≥ 100 µg/mL)	Not reported	[70]
Catharanthine (Figure 4g)	*Catharanthus roseus* (L.) G.Don	*P. aeruginosa* (MIC = 400 µg/mL)	Not reported	[71]
Conessine (Figure 4h)	*Holarrhena antidysenterica* (L.) Wall.ex A. DC.	*P. aeruginosa* (MIC = 40 µg/mL)	Not reported	[72]
Coumarins	*Mesua ferrea* L.	*S. aureus* (MIC = 3.12–100 µg/mL)	Not reported	[73]
Crysoplenol, Crysoplenetin	*Artemissia annua* L.	*S. aureus* (MIC = 250–500 µg/mL)	Not reported	[74]
Cucurbitane-type triterpenoids	*Momordica balsamina* L. (Cucurbitaceae)	*E. faecalis* (MIC = 100–>200 µg/mL), MRSA (MIC = 25–50 µg/mL)	Not reported	[75]
Cumin	*Cuminum cyminum* L.	*S. aureus* (MIC = 5–25 µg/mL)	Not reported	[76]
Epigallocatechin gallate (Figure 4i)	*Camellia sinensis* (L.) Kuntze	Not reported	MRSA (MIC ≤ 100 µg/mL)	[77]
Ferruginol (Figure 4j)	*Chamaecyparis lawsoniana* (A.Murray bis) Parl.	*S. aureus* (MIC = 4–128 µg/mL)	Not reported	[78]
Gallotannin glucopyranose	*Terminalia chebula* Retzius	*E. coli* (MIC = 12.1–97.5 µg/mL)	Not reported	[79]
Indirubin (Figure 4k)	*Wrightia tinctoria* (Roxb.) R. Br.	*S. epidermidis* (MIC = 25 µg/mL), *S. aureus* (MIC = 12.5 µg/mL)	Not reported	[80]
Kaempferol rhamnoside (Figure 4l)	*Persea lingue* (Ruiz & Pav.) Nees	*S. aureus* (IC_50_ = 2 µM)	Not reported	[81]
Oleanolic acid (Figure 4m)	*Carpobrotus edulis* (L.) L.Bolus	MRSA (MIC = 25–50 µg/mL)	Not reported	[82]
Osthol (Figure 4n)	*Cnidii monnieri* (L.)	*S. aureus* (MIC = 25 µg/mL), *P. aeruginosa* (MIC = 128 µg/mL)	Not reported	[83,84]
Phenylpropanoid ailanthoidiol	*Zanthoxylum capense* (Thunb.) Harv.	*S. aureus* (MIC = 50–100 µg/mL)	Not reported	[85]
Reserpine (Figure 4o)	*Rauwolfia vomitoria* Afzel., *Rauwolfia serpentina* (L.) Benth. ex Kurz	*E. coli* (ZOI = 15.5–16.5 mm)	Not reported	[86]
SB-202742 (Anacardic acid derivatives)	*Spondias mombin* L.	Not reported	*E. coli* (IC_50_ = 10.1 µg/mL), *P. aeruginosa* (IC_50_ = 40.5 µg/mL), *P. mirabilis* (IC_50_ = 111.3 µg/mL)	[87]
Silybin (Figure 4p)	*Silybum marianum* (L.) Gaertn	MRSA (ZOI = 29 mm)	Not reported	[88]
Tellimagrandin I	*Rosa canina* L.	Not reported	MRSA (MIC = 150 µg/mL)	[89]

**Table 3 ijms-26-05550-t003:** Synergistic interactions of phytochemicals in combination with clinical antibiotics.

Phytochemicals	Combination with Antibiotic	Bacterial Target	Inhibitory Activity Against	References
2,6-dimethyl-4-phenyl-pyridine-3,5-dicarboxylic acid diethyl ester	Ciprofloxacin	*S. aureus*	Efflux pump (MIC = 3–8 µg/mL)	[98]
5-O-Methylglovanon	Ampicillin	*S. aureus*, *S. epidermidis*	β-lactamase	[99]
α-pinene	Erythromycin, Tetracycline	*S. aureus*	Efflux pump (MIC = 128 µg/mL)	[100]
Baicalin	Cefotaxime, methicillin, benzylpenicillin, amoxicillin, ampicillin	MRSA	β-lactamase (MIC = 4–25 µg/mL)	[101]
Berberine	5ʹ- Methoxyhydnocarpin	*S. aureus*	Efflux pump (MIC = 32 µg/mL)	[102]
Budmunchiamines	Chloramphenicol	*E. coli*	Efflux pump (MIC = 8–64 µg/mL)	[103]
Carnosol, carnosic acid	Erythromycin	*S. aureus*	Efflux pump (MIC = 32–256 µg/mL)	[104]
Carvacrol	Erythromycin	Erythromycin-resistant Group A *Streptococci*	Efflux pump (MIC = 8–64 µg/mL)	[105]
Chanoclavine	Tetracycline	MDE *E. coli*	Efflux pump (MIC = 0.78–100 µg/mL)	[106]
Corosolic acid	Carbapenems	*E. coli*	β-lactamase (MIC = 0.08–1.5 µg/mL)	[107]
Curcumin, salicylate	Colistin	Carbapenem resistant *Enterobacteriaceae*	Efflux pump (MIC = 0.01–4 µg/mL)	[108]
Epigallocatechin-gallate	Tetracycline	*Staphylococcal* isolates	Efflux pump (MIC = 0.06–32 µg/mL)	[109]
Epigallocatechin-gallate	Sulbactam/ampicillin	MRSA	β-Lactamase (MIC = 4 µg/mL)	[110]
Gallic acid, quercetin, luteolin, protocatechuic acid	Amphenicol, tetracycline, fluoroquinolone, quinolone, β lactams	*E. coli*, *S. aureus*	β-Lactamase (MIC = 0.02–47 µg/mL) for *E. coli* and (MIC = 0.02–7.8 µg/mL) for S. aureus	[111]
Isoalantolactone	Penicillin G	*S. aureus*	β-Lactamase (MIC = 0.008–16 µg/mL)	[112]
Methyl-1alpha-acetoxy-7alpha 14alpha-dihydroxy-8,15-isopimaradien-18-oate, Methyl-1alpha,14alpha-diacetoxy-7alpha-hydroxy-8,15-isopimaradien-18-oate	Erythromycin, tetracycline	*S. aureus*	Efflux pump (MIC = 32–256 µg/mL)	[113]
Proanthocyanidins	Ampicillin, meropenem, cefotaxime	*E. coli*, *Staphylococci* strains, *Klebsiella*	β-Lactamase (MIC = 37.5–2400 µg/mL)	[114]
Quercetin	Meropenem	Carbapenem resistant *K. pneumoniae*, *E. coli*, *P. aeruginosa*, *A. baumannii*	Efflux pump	[115]
Tellimagrandin I, corilagin, ellagic acid, gallic acid	Oxacillin	MRSA	β-Lactamase (MIC = 0.25–256 µg/mL)	[89]
Tetrandrine	Colistin (Toxicity has been reported)	Colistin-resistant *Salmonella*	Efflux pump (MIC = 0.004–4 µg/mL)	[116,117]
Thymol, Carvacrol	Norfloxacin	*S. aureus*	Efflux pump (MIC = 32 µg/mL)	[118]

## Data Availability

Not applicable.

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
