# Peer review of "Plant Metabolites as Potential Agents That Potentiate or Block Resistance Mechanisms Involving β-Lactamases and Efflux Pumps"

_ijms, 2025, doi:10.3390/ijms26125550_

Round 1
Reviewer 1 Report
Comments and Suggestions for Authors
This review offers a comprehensive and timely discussion on natural products targeting β-lactamase and efflux pump resistance mechanisms. The topic is of significant relevance to the ongoing antimicrobial resistance (AMR) challenges, and the manuscript is well-structured, demonstrating a clear logical flow from the mechanisms to therapeutic potential. Tables and figures effectively summarize the key findings, and the inclusion of recent studies, such as those on KPC-2 and berberine, ensures the review is up to date. However, there are several major revisions needed to improve the quantitative analysis and deepen the insights.
1.The chemical structures, activities, and mechanisms of action of the natural products are not sufficiently detailed, which makes the review incomplete.
2.Figure 1, titled “Base structures of (a) phenolic acid, (b) coumarins, (c) flavonoids, (d) lignans, (e) stilbenes, (f) tannins, and (g) terpenoids,” features structures that are too large and are poorly arranged, leading to an inefficient layout.
3.In Tables 2 and 3, some compounds are only marked as “active,” but their IC50/MIC values are not provided, which hinders effective comparison of their potency.
4.It is recommended to include structure-activity relationships (SAR) for the major compound classes discussed.
5.The review does not address whether bacteria might develop resistance to natural inhibitors, such as rhein and quercetin.
6.The concept of "synergistic effects" is not standardized (e.g., there is no consistent use of the FIC index or dose reduction ratios).
Author Response
Reviewer 1
This review offers a comprehensive and timely discussion on natural products targeting β-lactamase and efflux pump resistance mechanisms. The topic is of significant relevance to the ongoing antimicrobial resistance (AMR) challenges, and the manuscript is well-structured, demonstrating a clear logical flow from the mechanisms to therapeutic potential. Tables and figures effectively summarize the key findings, and the inclusion of recent studies, such as those on KPC-2 and berberine, ensures the review is up to date.
We thank the reviewer for the positive comments.
1.The chemical structures, activities, and mechanisms of action of the natural products are not sufficiently detailed, which makes the review incomplete.
The reviewer has raised several points in this comment. We address each point separately below:
- With regards to the chemical structure, a new figure (Fig 4) has been added to the text in response to this comment (as well as a comment by another reviewer). This provides substantially more structural details about the potentiating phytochemicals.
- With regards to the activity part of this comment, Table 2 lists the activities in [efflux pump and β-lactamase activity] against specific bacteria and therefore we believe this comment has already been addressed. The activities of selected compounds have also been discussed in the text.
- In response to the comment regarding mechanisms, some of these are still under investigation, as stated in the manuscript. However, to address the known mechanisms of potentiation, we have added another section (6. Plant metabolite structure activity relationship).
This adds to the discussion of the mechanisms of action of natural products already included in the manuscript.
Additionally, we had already included several sections on the mechanisms in sections 8 and 9 for several well-characterised phytochemicals. For example, we draw the reviewer's attention to the following text”
- on page 2: “These enzymes exhibit shared structural characteristics and operate through a common mechanism of action. They variably hydrolyse cephalosporins, penicillins, carbapenems, monobactams, and are inhibited by tazobactam and clavulanate [17].”, which discusses that discuss efflux pump inhibiting activity of baicalin.
- In section 8, lines 78-132 from “Baicalin may disrupt ATP production, potentially inhibiting the function of the MsrA efflux pump and enhancing its efflux pump inhibitory activity. Studies have also shown capsaicin (Figure 4f) ….” To the end of section 8.
- Additionally, the β-lactamase inhibitory mechanisms are discussed in section 9, from “The phytochemical 2-methoxy chrysophanol (Figure 4c), derived from Clutia myricoides Jaub. & Spach possesses an anti-ESBL activity against K. pneumoniae. To investigate the mechanism of action of 2-methoxy chrysophanol, molecular docking was conducted with CTX-M-27 ESBL [62]…..” to the end of the section.
2.Figure 1, titled “Base structures of (a) phenolic acid, (b) coumarins, (c) flavonoids, (d) lignans, (e) stilbenes, (f) tannins, and (g) terpenoids,” features structures that are too large and are poorly arranged, leading to an inefficient layout.
To address the reviewer’s comment, the size of the structures has been reduced and the figure has been rearranged. It is now included in profile format rather than landscape format.
- In Tables 2 and 3, some compounds are only marked as “active,” but their IC50/MIC values are not provided, which hinders effective comparison of their potency.
We agree with the reviewer that IC50/MIC values should be provided for effective comparison. We have now added these values (where available) in Table 2 and 3.
- It is recommended to include structure-activity relationships (SAR) for the major compound classes discussed.
We agree with the reviewer that the inclusion of structure-activity relationships will add more depth to the paper. To address this point, we have added a new section, Section 7, “Plant metabolite structure activity relationship” that explores the structure and activity of plant metabolites. This has resulted in the addition of six references from 91-96.
5.The review does not address whether bacteria might develop resistance to natural inhibitors, such as rhein and quercetin.
In response to the reviewer comments the following text has now been added to lines 222-226:
“Whilst all of the combinational therapies summarised herein highlight the potential to overcome bacterial resistance mechanisms. It is noteworthy that the bacteria may develop resistance against the potentiating compound, and further work is required to address this”.
- The concept of "synergistic effects" is not standardized (e.g., there is no consistent use of the FIC index or dose reduction ratios).
We agree with the reviewer and the following text has now been added to 213-214:
“The concept of synergistic effects is not standardised due to differences between the studies, and there is no uniform use of the term across studies.”
Reviewer 2 Report
Comments and Suggestions for Authors
In this study, the authors have reviewed different plant metabolites as inhibitors of the two most pronounced mechanisms of bacterial resistance, the production of beta-lactamases and the efflux pumps’ activity, to the clinically relevant antibiotics.
Addressing the current state of bacterial resistance to the clinically used antibiotics, especially the beta-lactams, is essential. Nevertheless, the study would be further strengthened if the authors addressed the following points:
- Title: Since the review is focused on plant metabolites, the authors might consider reflecting that focus in the title, e.g., instead of “Natural products…”, “Plant metabolites…” or “Phytochemicals as inhibitors of the bacterial efflux pumps and beta-lactamases”.
- Somehow, the lines in the pages before the subtitle 6. “Synergistic interactions between efflux pump and β-lactamase inhibitor phytochemicals and conventional antibiotics” are missing. The line numbering starts on page 8.
- Overall, the chemical structures presented in the manuscript will benefit from using a professional drawing program, such as ChemDraw. There are trial periods offered by companies that the authors could make use of.
- Figures: In Figure 3, the bond between the carbonyl group and the hydroxyl group of the carboxylic acid (in the structure to the left) is missing. Figure 5 should be placed earlier in the text, after the subtitle 6. “Synergistic interactions between efflux pump and β-lactamase inhibitor phytochemicals and conventional antibiotics”, and before Table 3 – line 18. In the paragraph starting (currently) from line 1, ending on line 17, clavulanic acid is mentioned, which is the clinically relevant inhibitor to date. Therefore, Fig. 5 should be introduced there.
- The structures of at least the plant metabolites discussed in the text, e.g., berberine, 2-methoxy chrysophanol, 1,4-naphthalenedione, epigallocatechin, quercetin, pigallocatechin gallate, etc., should be incorporated in the text. This will allow the reader to see the similarities in the plant metabolites’ structures with a similar mode of action. The need for structural representation is emphasized when a given compound is represented by its chemical name, e.g., 2-methoxy chrysophanol, 1,4-naphthalenedione, and the binding to its target is known and discussed by the authors in detail (see lines 121-139).
- There are somewhat contradictory statements in subtitle 9. “Challenges and Future Perspectives”, the sentence starting on line 171 “Isolating individual phytochemicals with targeted antibacterial activity can be a lengthy process and may require the use of large quantities of plant material and extraction solvents that can be toxic or environmentally damaging”, and the sentence starting on line 189 “The structures of these compounds should also be modified to optimise their pharmacodynamics and pharmacokinetics, alongside conducting structure-activity analysis to enhance their safety and efficacy”. The authors should clarify how one modifies the structure without isolating it from the mixture of compounds, which is always the case with natural products.
Author Response
Reviewer 2
- Title: Since the review is focused on plant metabolites, the authors might consider reflecting that focus in the title, e.g., instead of “Natural products…”, “Plant metabolites…” or “Phytochemicals as inhibitors of the bacterial efflux pumps and beta-lactamases”.
We agree with this comment. The title has now been changed to “Plant metabolites as potential agents that potentiate or block resistance mechanisms involving β-lactamases and efflux pumps”
- Somehow, the lines in the pages before the subtitle 6. “Synergistic interactions between efflux pump and β-lactamase inhibitor phytochemicals and conventional antibiotics” are missing. The line numbering starts on page 8.
This is because inserting the tables into the text terminated the line numbering. As these line numbers are not included in the published version of the manuscript, this is only an issue for reference to the reviewer comments. We would change this by removing the figures and tables, renumbering the lines and then reinserting the figures and tables, but that would show nearly the entire manuscript as changes, making it difficult to locate the revisions made. So that the reviewer can see the revisions, we have left this in its current form.
- Overall, the chemical structures presented in the manuscript will benefit from using a professional drawing program, such as ChemDraw. There are trial periods offered by companies that the authors could make use of.
We agree with the reviewer’s comment and have now revised the chemical structures throughout the manuscript.
- Figures: In Figure 3, the bond between the carbonyl group and the hydroxyl group of the carboxylic acid (in the structure to the left) is missing.
This has now been corrected in the revised figure.
Figure 5 should be placed earlier in the text, after the subtitle 6. “Synergistic interactions between efflux pump and β-lactamase inhibitor phytochemicals and conventional antibiotics”, and before Table 3 – line 18.
We disagree with the reviewer on this point. The section 6 (in the original manuscript) that the reviewer is referring to is a general introduction to synergistic interactions, which does not go into the mechanism. Instead, we believe that a figure showing β lactamase inhibition mechanism is better located in the section that discusses this mechanism i.e. section 9 β-Lactamase inhibitor mechanism (in the revised manuscript).
In the paragraph starting (currently) from line 1, ending on line 17, clavulanic acid is mentioned, which is the clinically relevant inhibitor to date. Therefore, Fig. 5 should be introduced there.
Whilst we agree with the reviewer that clavulanic acid is mentioned in section 6, we still believe that section 9 (in the revised manuscript) is a better location for this figure as the figure shows the mechanism of inhibition, which is what section 9 is about.
- The structures of at least the plant metabolites discussed in the text, e.g., berberine, 2-methoxy chrysophanol, 1,4-naphthalenedione, epigallocatechin, quercetin, epigallocatechin gallate, etc., should be incorporated in the text. This will allow the reader to see the similarities in the plant metabolites’ structures with a similar mode of action. The need for structural representation is emphasized when a given compound is represented by its chemical name, e.g., 2-methoxy chrysophanol, 1,4-naphthalenedione, and the binding to its target is known and discussed by the authors in detail (see lines 121-139).
We agree with the reviewer that the structure of some of the plant metabolites discussed in the paper should be incorporated in the text. We have added Figure 4 in the text that shows the structures of:
Figure 4: (a) Berberine, (b) 1, 4-naphthalenedione, (c) 2-methoxy chrysophanol (d) aranorosin, (e) baicalein, (f) capsaicin, (g) catharanthine, (h) conessine, (i) epigallocatechin gallate, (j) ferruginol, (k) indirubin, (l) kaempferol rhamnoside, (m) oleanolic acid, (n) osthol, (o) reserpine, and (p) silybin.
- There are somewhat contradictory statements in subtitle 9. “Challenges and Future Perspectives”, the sentence starting on line 171 “Isolating individual phytochemicals with targeted antibacterial activity can be a lengthy process and may require the use of large quantities of plant material and extraction solvents that can be toxic or environmentally damaging”, and the sentence starting on line 189 “The structures of these compounds should also be modified to optimise their pharmacodynamics and pharmacokinetics, alongside conducting structure-activity analysis to enhance their safety and efficacy”. The authors should clarify how one modifies the structure without isolating it from the mixture of compounds, which is always the case with natural products.
We disagree with the reviewer that the sentences starting from line 171 (in the original manuscript), which is now several lines later due to the addition of the text and line 189 (in the revised manuscript).
- The first of these sentences: “Isolating individual phytochemicals with targeted antibacterial can be a lengthy process and may require the use of large quantities of plant material and extraction solvents that can be toxic or environmentally damaging” highlights one of the challenges associated with the use of natural products for therapeutic use.
- The second of these sentences: “The structure of these compounds should also be modified to optimise their pharmacodynamics and pharmacokinetics, alongside conducting structure-activity analysis to enhance their safety and efficacy” focuses on future perspective about the use of natural products.
Please note that both of these sentences are part of the section 10 challenges and future perspectives, and whilst both relate to challenges in this field, they are distinct statements and therefore do not contradict each other.